# Accuracy of intraocular lens power calculation formulas using a swept-source optical biometer

**Se Young Kim, Seung Hyun Lee, Na Rae Kim, Hee Seung Chin, Ji Won Jung** *

Department of Ophthalmology and Inha Vision Science Laboratory, Inha University School of Medicine, Incheon, South Korea

* panch325@gmail.com

## Abstract

### Purpose

To compare the accuracy of the five commonly used intraocular lens (IOL) calculation formulas integrated to a swept-source optical biometer, the IOLMaster 700, and evaluate the extent of bias within each formula for different ocular biometric measurements.

### Methods

The study included patients undergoing cataract surgery with a ZCB00 IOL implant, using IOLMaster 700 optical biometry. A single eye per patient was included in the final analysis for a total of 324 cases. The SRK/T, Hoffer Q, Haigis, Holladay 2, and Barrett Universal II formulas were evaluated. The correlations between the refractive prediction errors calculated using the five formulas and ocular dimensions such as axial length (AL), anterior chamber depth (ACD), corneal power, and lens thickness (LT) were analyzed.

### Results

There were significant differences in the median absolute error predicted by the five formulas after the adjustment for mean refractive prediction errors to zero ($P = 0.038$). The Barrett Universal II formula had the lowest median absolute error (0.263) and resulted in a higher percentage of eyes with prediction errors within ±0.50 D, ±0.75 D, and ±1.00 D (all $P < 0.050$). The refractive errors predicted by only the Barrett formula showed no significant correlation with the ocular dimensions: AL, ACD, corneal power, and LT.

### Conclusions

Overall, the Barrett Universal II formula, integrated to a swept-source optical biometer had the lowest prediction error and appeared to have the least bias for different ocular biometric measurements for the ZCB00 IOL.

**Data Availability Statement:** All relevant data are within the manuscript.

**Funding:** This work was supported by Basic Science Research Program through the National Research Foundation of Korea (NRF) funded by the

Ministry of Education (2017R1D1A1B03034469) and by INHA UNIVERSITY Research Grant (60195-01). The funders had no role in study design, data collection and analysis, decision to publish, or preparation of the manuscript.

**Competing interests:** The authors have declared that no competing interests exist.

## Introduction

The development of optical biometry and intraocular lens (IOL) power calculation formulas has improved the refractive outcomes of cataract surgery. Advanced technologies related to optical biometry such as partial coherence interferometry (PCI), optical low-coherence reflectometry (OLCR), and swept-source optical coherence tomography (SS-OCT) have increased the precision of biometric measurements. [1–4] Modern IOL power calculation formulas have tried to improve the accuracy of their predictions of effective lens position (ELP). For the most part, this has been accomplished by increasing the number of variables—including preoperative anterior chamber depth (ACD, measured from epithelium to lens), lens thickness (LT), corneal diameter, preoperative refraction, and age—as well as basic variables such as axial length (AL) and corneal power (K).

The IOL calculation formulas show similarly accurate refractive results in eyes with normal AL. [5] However, the accuracy of these formulas differ in eyes with short and long AL. [5–7] The Hoffer Q formula provide the more accurate outcomes in eyes with a short AL [5,8,9] and the SRK/T and Haigis formulas are suitable in eyes with a long AL. [8,10–13] Nevertheless, accurately predicting the ELP remains a major source of error in IOL power calculations, and controversy persists about the accuracy of refractive predictions among many formulas. [14] Because there is no single highly accurate formula across a range of eye characteristics such as long or short AL, flat or steep cornea, and deep or shallow ACD, many cataract surgeons should consider and use several formulas in eyes with various ocular dimensions. [8,15,16]

The Barrett Universal II formula was recently introduced and its accuracy has been studied, and better refractive outcomes than those of other formulas have been reported. [14,16,17–19] The newly developed IOLMaster 700 (Carl Zeiss Meditec AG, Jena, Germany) adopted SS-OCT technology and recently integrated the latest-generation Barrett IOL power calculation formulas. Therefore, cataract surgeons can automatically apply this formula using this device.

The purpose of this study was to determine which of the commonly used IOL formulas integrated to the IOLMaster 700 swept-source optical biometer is the best predictor of actual postoperative refractive outcomes: SRK/T, Hoffer Q, Haigis, Holladay 2, and Barrett Universal II. We also evaluated the extent of bias within each formula for the different ocular biometric measurements (AL, corneal power, ACD, LT).

## Materials and methods

This retrospective chart review comprised all cataract surgeries performed in 2018 and 2019 at a tertiary center. The study received approval from the institutional review board of Inha University Hospital (no. 2018-11-010), and the IRB waived the requirement for informed consent. All research and data collection followed the tenets of the Declaration of Helsinki. Confidentiality of the information was maintained thoroughly by excluding names as identification in data abstraction form and keeping their privacy during data collection. No one had access to the non-coded data except investigators, data collectors and supervisor due to responsibilities associated with the study. This retrospective cross-sectional study included consecutive Korean patients who underwent uncomplicated phacoemulsification with an implantation of the most commonly used IOL (TECNIS® ZCB00, Johnson & Johnson Vision, Santa Ana, CA, USA) at our institution. Two surgeons performed the surgery by clear corneal temporal incision phacoemulsification. All patients underwent preoperative measurements by the IOLMaster 700, a swept-source optical biometer.

Our selection criteria for the study subjects and methods followed the recommendations of recent studies regarding the protocols for studies of IOL formula accuracy. [20,21] The

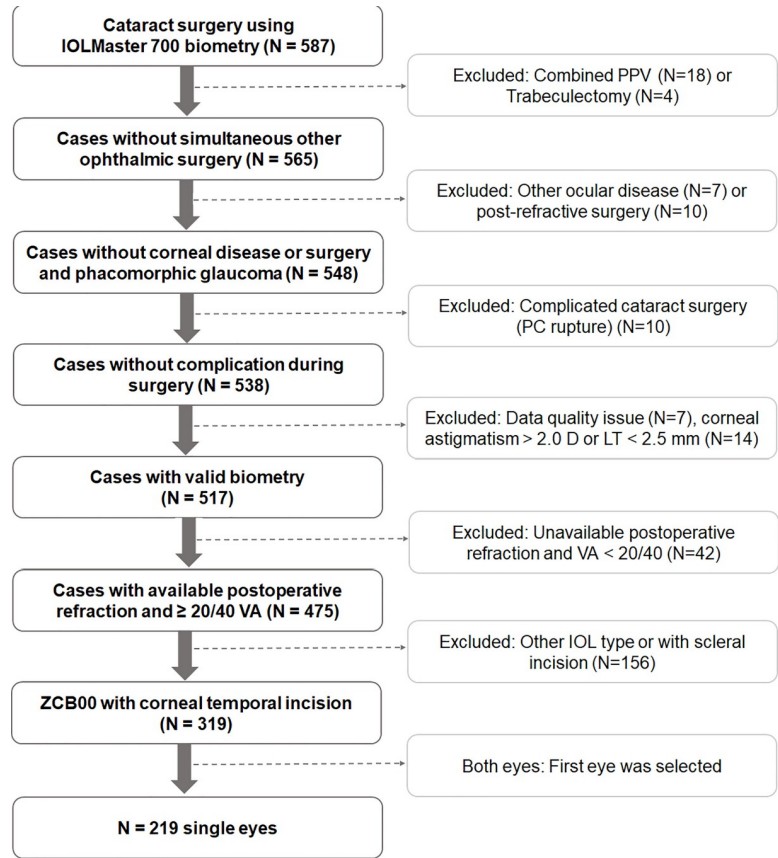

**Fig 1. Overview of the study selection process.**

exclusion criteria were incomplete biometry, corneal astigmatism more than 2.0 diopters (D), LT measurement less than 2.50 mm, complicated cataract surgery (posterior capsular rupture), additional procedures during cataract surgery (combined vitrectomy or glaucoma surgery), postoperative corrected distance visual acuity (CDVA) worse than 20/40, refraction performed before 4 weeks postoperatively, postoperative complications, and incomplete documentation. Patients with a history of corneal disease or refractive surgery and phacomorphic glaucoma were excluded. If both eyes were eligible, the first eye was selected. Fig 1 shows an overview of the study's selection criteria.

The commonly used and more recent five IOL power calculation formulas built-in software of IOLMaster 700 (software version 1.8) were evaluated: SRK/T, Hoffer Q, Haigis, Holladay 2, and Barrett Universal II. Lens constant optimizations for the ZCB00 IOL were performed in collaboration with Carl Zeiss Meditec AG, which has licensed versions of the proprietary Barrett Universal II and Holladay 2 as well as implementations of the SRK/T, Hoffer Q, and Haigis formulas. [5, 22, 23] The A-constant for SRK/T was 119.3 and the pseudophakic ACD was 5.80 for the Hoffer Q formula. The a0, a1, and a2 constants were -1.302, 0.210, and 0.251, respectively, for the Haigis formula and the ACD was 5.786 for the Holladay 2. The lens factor was 2.04 for the Barrett Universal II. [24]

Postoperative subjective manifest refraction was measured at least 1 month after surgery, when the refraction is considered stable. The refractive prediction error was then calculated as the actual postoperative refraction minus the refractive result predicted by each formula for the IOL implanted. The mean refractive prediction errors for each formula were zeroed out by

adjusting the refractive prediction error for each eye. After the adjustment of the mean refractive prediction error to zero, the standard deviation (SD) of prediction error, median absolute error (MedAE), and mean absolute error (MAE) for each formula were calculated. The percentages of eyes within ±0.25 D, ±0.50 D, ±0.75 D, and ±1.00 D of the refractive prediction error were calculated.

### Statistical analysis

All statistical analyses were performed using SPSS for Windows (version 20.0; SPSS Inc., Chicago, IL, USA). To compare the accuracies of the five formulas, we used the Friedman nonparametric test of the MedAE. The post-hoc test of the Wilcoxon signed rank test was performed for multiple comparisons of the formulas. We used Cochran's Q test to compare the percentage of eyes within a certain range of prediction errors between the five formulas. The post-hoc test of McNemar's test was performed for multiple comparisons of the formulas. Bonferroni correction was applied for multiple comparisons. Linear regression analysis was used to evaluate the correlation between refractive errors predicted by each formula and preoperative biometric factor. Adjusted $P$ values (by Bonferroni correction) less than 0.05 were considered statistically significant.

## Results

Data from 324 eyes of 324 patients were evaluated. The majority (n = 179, 55.2%) of the samples were left eyes and more women (n = 193, 59.6%) than men underwent cataract surgery during the study period. The demographic and biometric characteristics of the patient populations are shown in Table 1. The mean axial length was 23.34 ± 1.10 mm, mean corneal power was 44.42 ± 1.65 diopter, and mean ACD was 3.06 ± 0.48 mm.

Table 2 shows the mean refractive prediction errors, SD of prediction error, MedAE, and MAE determined by the five formulas in the 324 eyes after the prediction errors for each formula were zeroed out. The MedAEs with adjusting the refractive prediction error to zero are shown in Fig 2. The Friedman test confirmed that there were statistically significant differences among the absolute prediction errors of the five formulas ($P$ = 0.038). Post hoc analysis using Wilcoxon signed-rank pairwise comparisons for nonparametric samples with Bonferroni correction showed that the Barrett had a significantly smaller MedAE than the other formulas; SRK/T ($P$ = 0.020), Hoffer Q ($P$ = 0.048), Haigis ($P$ = 0.012), and Holladay 2 ($P$ = 0.024).

The percentage of eyes within a certain range of prediction errors is shown in Table 2 and Fig 3. The percentages of eyes within ±0.50 D, ±0.75 D, and ±1.00 D of error were significantly different among the five formulas using the Cochran's Q test (all $P$ < 0.050). Post hoc analysis using McNemar's test with Bonferroni correction was performed. The Barrett formula produced a higher percentage of eyes within ± 0.50 D of error than the Hoffer Q and Holladay 2 formulas ($P$ < 0.001 and $P$ = 0.016). The Barrett formula produced a higher percentage of eyes within ± 0.75 D of error than the other formulas; SRK/T ($P$ = 0.048), Hoffer Q ($P$ = 0.008), Haigis ($P$ = 0.020), and Holladay 2 ($P$ = 0.004). The Barrett also produced a higher percentage of eyes within ± 1.00 D of error than the Holladay 2 ($P$ = 0.016).

The refractive errors predicted by all formulas except that by the Barrett Universal II were significantly correlated with the AL on linear regression analysis (all $P$ < 0.050). The refractive errors predicted by the SRK/T formula showed a significant negative correlation with keratometry ($P$ < 0.001), while the Hoffer Q and Haigis formula showed a significant positive correlation with keratometry ($P$ = 0.023 and $P$ < 0.001). The Hoffer Q and Haigis formulas showed a significant positive correlation with ACD ($P$ < 0.001 and $P$ = 0.027), and the refractive errors predicted by the Haigis and Holladay 2 formulas were correlated with LT (all $P$ < 0.001; Fig 4).

**Table 1. Demographics and biometric data using a single optical biometry device (IOLMaster 700) in the patients who underwent cataract surgery (n = 324).**

| Parameter | Mean ± SD | Range |
|---|---|---|
| Age (years) | 69.6 ± 9.9 | 39–90 |
| AL (mm) | 23.34 ± 1.10 | 20.93–27.35 |
| Km (D) | 44.42 ± 1.65 | 40.13–48.12 |
| ACD (mm) | 3.06 ± 0.48 | 2.02–4.66 |
| LT (mm) | 4.47 ± 0.45 | 2.70–5.42 |
| IOL power (D) | 21.56 ± 2.77 | 10.00–32.00 |
| | Count (% of total) | |
| AL subgroups | | |
| Short (<22.0 mm) | 22 (6.8%) | |
| Medium (22.0–26.0 mm) | 296 (98.1%) | |
| Long (>26.0 mm) | 6 (1.9%) | |
| Keratometry subgroups | | |
| Flat (<42.0 D) | 23 (7.1%) | |
| Medium (42.0–46.0 D) | 244 (75.3%) | |
| Steep (>46.0 D) | 57 (17.6%) | |
| ACD subgroups | | |
| Shallow (<2.5 mm) | 38 (11.7%) | |
| Medium (2.5–3.5 mm) | 222 (68.5%) | |
| Deep (>3.5 mm) | 64 (19.8%) | |

AL, axial length; ACD, anterior chamber depth; Km, mean keratometry; LT, lens thickness; IOL, intraocular lens

## Discussion

To our knowledge, this is the first study to compare the accuracy of IOL power calculation formulas on one IOL type (TECNIS® ZCB00) using a swept-source optical biometer, the IOLMaster 700. We reported on five commonly used IOL calculation formulas: popular third-generation (SRK/T, Hoffer Q) and fourth-generation (Haigis, Holladay 2, and Barrett Universal II). These formulas were preinstalled on the IOLMaster 700. We followed the recently published protocols comparing their respective accuracies. [20,21]

Overall, the refractive outcomes and percentages of eyes with prediction errors within ±0.25 D, ±0.50 D, ±0.75 D, and ±1.00 D for each formula were similar to those in the recent study by Melles et al. using a Lenstar 900 optical biometer. [16] All five formulas achieved above 92% of eyes within ±1.00 D of the predicted refraction, much higher than the 85% suggested by Gale et al. [25] Recent studies reported that the Barrett Universal II formula was more accurate and showed the better refractive outcomes than the other formulas. [16,17–19,26] One large population study assessed the Barrett Universal II formula over the entire AL range and showed that this formula had the lowest MAE and SD of the prediction error and a higher percentage of eyes with prediction errors within ±0.25 D, ±0.50 D, and ±1.00 D, which was congruent with our findings. [18] In our study, the Barrett Universal II formula had the lowest median absolute error (0.263) and a higher percentage of eyes with prediction errors within ±0.50 D, ±0.75 D, and ±1.00 D compared to the other formulas. Cooke and Cooke [17] found that the same formula could give different results depending on the optical biometer (OLCR and PCI) and the preinstalled version or not. Our results suggested that the Barrett Universal II formula was the most accurate among the commonly used and representative five

**Table 2. Clinical outcomes of refractive prediction error and absolute error and among the five IOL formulas after adjusting the mean refractive prediction error to zero (n = 324).**

| Formula | Mean RE | SD | MedAE | MAE | Percentage of eyes within diopter range indicated (%) | | | | |
|---|---|---|---|---|---|---|---|---|---|
| | | | | | ±0.25D | ±0.50D | ±0.75D | ±1.00D | >±2.00D |
| SRK/T | 0.000 | 0.472 | 0.310 | 0.376 | 41.0% | 73.9% | 87.3% | 96.3% | 0% |
| Hoffer Q | 0.000 | 0.520 | 0.290 | 0.396 | 44.8% | 67.8% | 86.1% | 94.3% | 0% |
| Haigis | 0.000 | 0.512 | 0.314 | 0.394 | 38.6% | 72.6% | 86.6% | 94.4% | 0% |
| Holladay 2 | 0.000 | 0.518 | 0.316 | 0.390 | 43.5% | 72.4% | 87.0% | 92.2% | 0% |
| Barrett Universal II | 0.000 | 0.426 | 0.263 | 0.334 | 42.7% | 79.4% | 92.4% | 97.2% | 0% |

SRK/T, Sanders-Retzlaff-Kraff/Theoretical; RE, refractive prediction errors; SD, standard deviation; MAE, mean absolute error; MedAE, median absolute error

formulas integrated to an advanced swept-source optical biometer in our study subjects, who had mostly normal ranges of ocular dimension.

Hoffer et al. [27] reported a similar accuracy of IOL power calculation using the Hoffer Q, Holladay 1, and SRK/T formulas using both SS-OCT and OLCR instruments. In their study, the MedAEs and the percentage of eyes with prediction errors within ±0.50 D for Hoffer Q using IOLMaster 700, were better than our results. They evaluated the outcomes of different IOL models (MX60 and SA60AT) and ocular dimensions of their subjects were relatively different from those of our subjects. These were estimated as the possible causes of this discrepancy.

Here we also evaluated the extent of bias within each formula for different ocular biometric measurements. The refractive errors predicted by all but the Barrett Universal II formula, was significantly correlated with the AL. The SRK/T and Haigis formulas have significant bias with varying corneal power in opposite directions. According to the ACD, Hoffer Q and Haigis formulas showed a significant positive correlation, and the refractive errors predicted by Haigis and Holladay 2 formulas were correlated with the LT. Overall, the Barrett formula appeared to have the least bias of the formulas as measured by prediction error with variations in AL, corneal power, ACD, and LT. These results were similar to those reported by Melles et al., [16] who found notable biases in the errors of all other formulas except the Barrett when plotted versus ocular dimensions using a Lenstar 900 biometer.

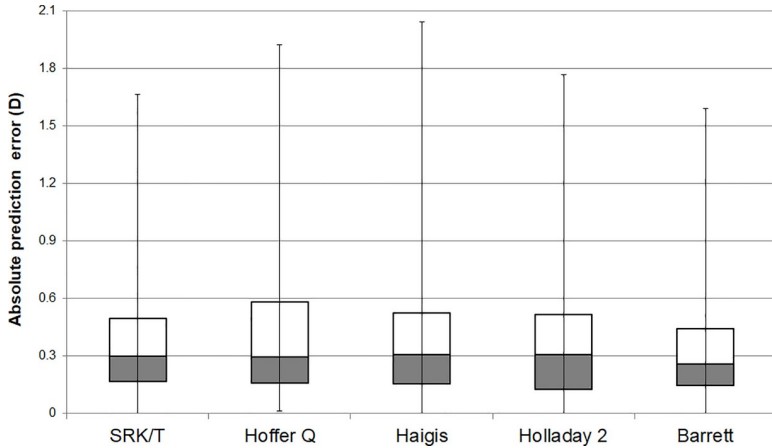

**Fig 2. Box plot of the absolute error (in diopters) of the five intraocular lens (IOL) calculation formulas for the ZCB00 model.** Dark gray boxes represent the second quartile, and white boxes represent the third quartile.

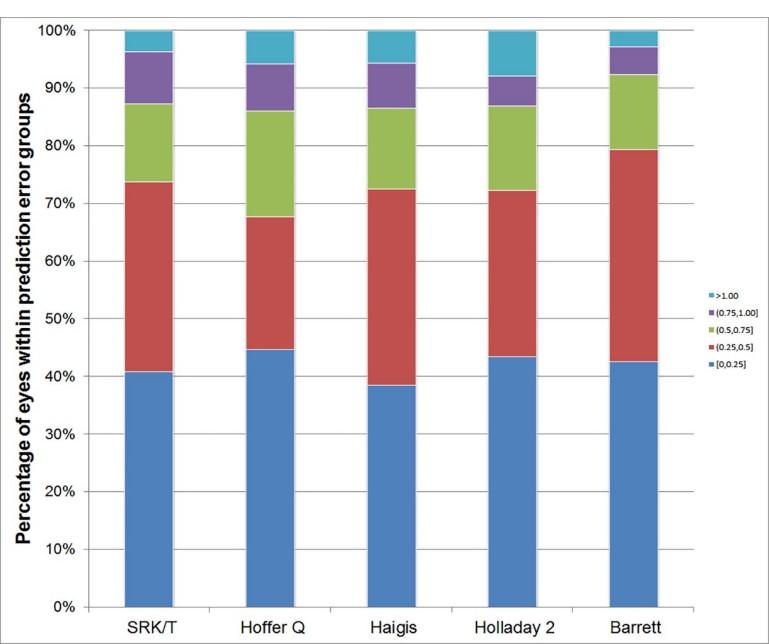

**Fig 3. Stacked histogram comparing the percentage of cases within a given diopter range of predicted refraction outcome of the five intraocular lens (IOL) calculation formulas for the ZCB00 model.**

When cataract surgeons select the IOL power during cataract surgery, they mainly use the preferred formula such as the Hoffer Q or SRK/T because modern IOL formulas have similar

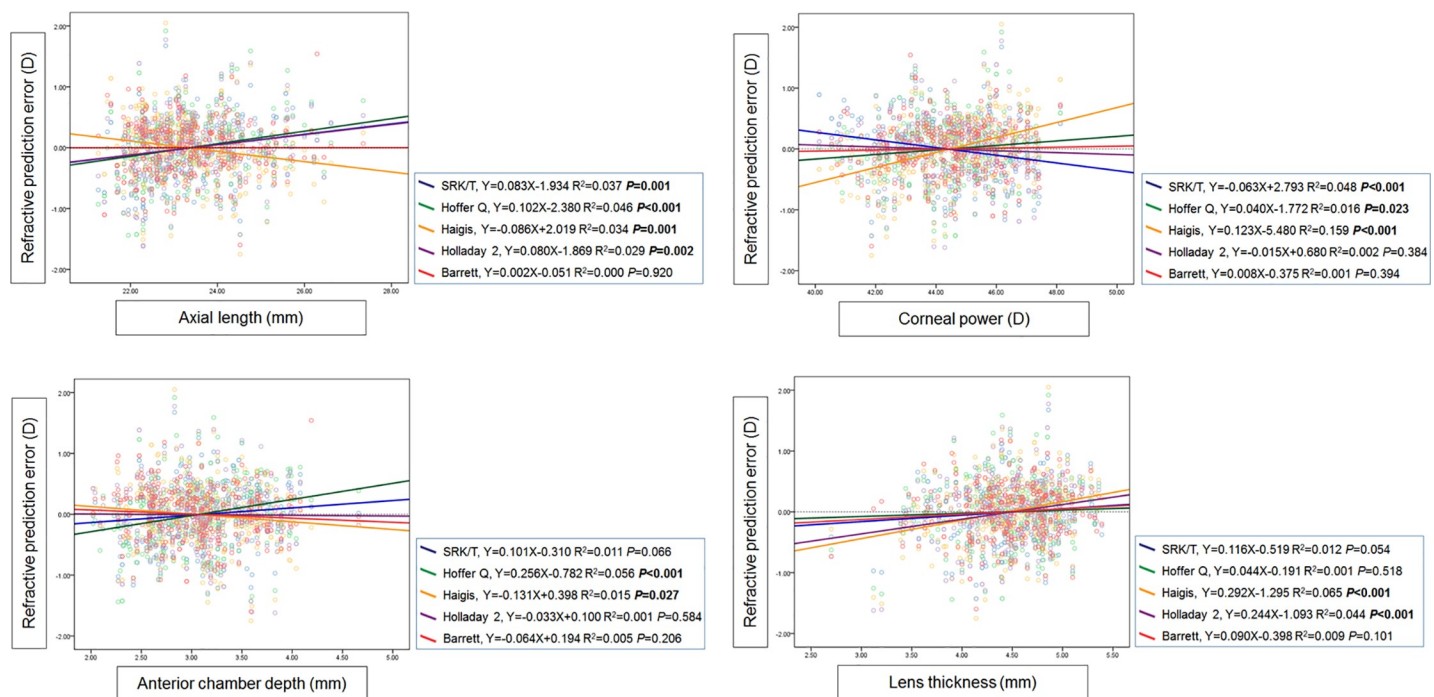

**Fig 4. Scatterplots showing the correlations between the refractive prediction error calculated using the five formulas and ocular dimensions including axial length (AL), anterior chamber depth (ACD), mean keratometry, and lens thickness (LT).**

accuracy in eyes with a normal range. [6,28] However, surgeons should cross-check different IOL formulas in eyes with an unusual range of ocular dimensions such as a short or long AL, flat or steep cornea, or a recently shallow ACD. A swept-source optical biometer, the IOLMaster 700, integrated the various IOL formulas including the latest-generation Barrett IOL power calculation formula, and we can automatically apply these formulas and compare the predicted results without using a separate program. Because the formulas gave different results depending on which optical biometry measurements were used and the preinstalled version, [17] we first compared the accuracy of various IOL formulas integrated to a device and confirmed the bias of these formulas as measured by prediction error with variations in ocular dimensions.

The present study has some limitations. First, because we evaluated one popular IOL model, we caution that these results may not be generalizable to other IOL models. Second, the sample size of eyes with unusual ranges of ocular dimensions was relatively small; therefore, further studies with larger sample size are needed in these subgroups.

In conclusion, we found statistically significant differences in the MedAEs for the five formulas after the adjustment for mean refractive prediction errors to zero. Overall, the Barrett Universal II formula, integrated to a swept-source optical biometer, had the lowest prediction error for ZCB00 IOL model. The Barrett Universal II formula also appeared to have the least bias as measured by prediction error with variations in different biometric ocular dimensions including AL, corneal power, ACD, and LT.

## Author Contributions

**Conceptualization:** Se Young Kim, Na Rae Kim, Hee Seung Chin, Ji Won Jung.

**Data curation:** Se Young Kim, Seung Hyun Lee, Ji Won Jung.

**Formal analysis:** Se Young Kim, Seung Hyun Lee, Na Rae Kim, Hee Seung Chin, Ji Won Jung.

**Investigation:** Ji Won Jung.

**Methodology:** Na Rae Kim, Hee Seung Chin, Ji Won Jung.

**Project administration:** Ji Won Jung.

**Resources:** Ji Won Jung.

**Software:** Ji Won Jung.

**Supervision:** Hee Seung Chin, Ji Won Jung.

**Validation:** Ji Won Jung.

**Visualization:** Ji Won Jung.

**Writing – original draft:** Se Young Kim, Ji Won Jung.

**Writing – review & editing:** Ji Won Jung.

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
