## [Decision Letter · Decision Letter 0]

16 Aug 2019

PONE-D-19-21173

Accuracy of intraocular lens power calculation formulas using a swept-source optical biometer

PLOS ONE

Dear Dr Jung,

Thank you for submitting your manuscript to PLOS ONE. After careful consideration, we feel that it has merit but does not fully meet PLOS ONE’s publication criteria as it currently stands. Therefore, we invite you to submit a revised version of the manuscript that addresses the points raised during the review process.

We would appreciate receiving your revised manuscript by Sep 30 2019 11:59PM. To enhance the reproducibility of your results, we recommend that if applicable you deposit your laboratory protocols in protocols.io, where a protocol can be assigned its own identifier (DOI) such that it can be cited independently in the future. For instructions see: http://journals.plos.org/plosone/s/submission-guidelines#loc-laboratory-protocols

We look forward to receiving your revised manuscript.

Kind regards,

Ireneusz Grulkowski, PhD

Academic Editor

PLOS ONE

Journal Requirements:

1. In ethics statement in the manuscript and in the online submission form, please provide additional information about the patient records used in your retrospective study. Specifically, please ensure that you have discussed whether all data were fully anonymized before you accessed them and/or whether the IRB or ethics committee waived the requirement for informed consent. If patients provided informed written consent to have data from their medical records used in research, please include this information.

2. Thank you for including your ethics statement in the manuscript: "The study received institutional review board approval (no. 2018-11-010), and all research and data collection adhered to the tenets of the Declaration of Helsinki."

4. Thank you for including the following funding information within your acknowledgements section; "This work was supported by Basic Science Research Program through the National Research Foundation of Korea (NRF) funded by the Ministry of Education (2017R1D1A1B03034469)."

"No"

Additional Editor Comments (if provided):

Please, address the issues raised by the reviewers.

Reviewers' comments:

Reviewer's Responses to Questions

**Comments to the Author**

1. Is the manuscript technically sound, and do the data support the conclusions?

Reviewer #1: Yes

Reviewer #2: Yes

Reviewer #3: Yes

Reviewer #4: Yes

2. Has the statistical analysis been performed appropriately and rigorously? 

Reviewer #1: No

Reviewer #2: Yes

Reviewer #3: Yes

Reviewer #4: Yes

3. Have the authors made all data underlying the findings in their manuscript fully available?

Reviewer #1: Yes

Reviewer #2: Yes

Reviewer #3: No

Reviewer #4: Yes

4. Is the manuscript presented in an intelligible fashion and written in standard English?

Reviewer #1: Yes

Reviewer #2: Yes

Reviewer #3: Yes

Reviewer #4: Yes

5. Review Comments to the Author

Reviewer #1: This is an interesting study. Although it does not add significant information to the recently published studies on much larger samples, it is the one of the first multiformula comparisons carried out with measurements by the IOLMaster 700.

Please find below some suggestions to improve the paper.

METHODS

• Subgroup analysis: short eyes are usually defined as those with AL <22.0 mm (not 22.5 mm). Please correct and repeat analysis.

• Statistical analysis: in addition to Friedman’s test (to analyze the absolute prediction errors), I would include Repeated Measures ANOVA to compare the arithmetical prediction errors and their variance. This can be useful for both the whole sample and the different subgroups (shallow, medium and deep ACD, etc.), where it may reveal statistically significant differences among the 5 formulas.

• Statistical analysis: since you correctly used it, please cite Cochran’s Q test in the Methods section too.

• Please include errata for the Hoffer Q formula in the references (J Cataract Refract Surg 1993;19:700-712; errata, 1994;20:677; 2007;33:2-3.)

RESULTS

• It is quite hard to believe the Cochran’s Q test was not able to detect any statistically significant difference among the 5 formulas, as – for example – the Hoffer Q achieved only 69.6% of eyes within 0.5D, whereas the Barrett achieved 78.1% with the same prediction error. Please check carefully the statistical analysis once more.

DISCUSSION

• Barrett formula does not belong to 5th generation formulas. Although unpublished and thus largely unknown, it should be classified among 4th generation ones.

• Please compare your results with those reported by Hoffer Q when comparing the IOLMaster 700 to the Lenstar (JCRS 2016;42:1165-1172). They included the Hoffer Q, Holladay 1 and SRK/T formulas and achieved better results. It would be opportune to elaborate on the possible causes of this discrepancy.

• The whole discussion should be reviewed after ANOVA has been applied to the arithmetic prediction errors, as statistically significant differences may arise.

Reviewer #2: Good results that support the conclusion. Reasonable statistical analysis. Correct terminology used. The ideas are clearly expressed in English. A bigger sample size is suggested in future research.

Reviewer #3: I have read the article by Kim et al. The authors have analyzed the accuracy of IOL power calculation formulas using the IOL Master 700.

Comments:

- P3 L53 - what about LCOR?

- P3 L56 - some formulas also use preoperative refraction and age. Moreover, you should state that increasing the number, from the basic which is AL and K

- P3 L63 - be more precise, what formulas in what axial lengths? Is it concordant with the results of your study?

- P5 L108 - short eyes were determined as those having AL<22.5 mm, while long as those having 25.0 mm. As several studies used different cut-off values i.e. 22.0 mm or 26.0 mm choosing these values should be thoroughly explained.

- the group size for some subgroups (e.g. long eyes=16, flat keratometry=16, shallow ACD=20) very small, making an appropriate analysis difficult.

- in the results section you state: "The percentages of eyes within the prediction errors of ±0.25, ±0.50, ±0.75 D, and Â±1.00 D were not significantly different among the five formulas using the Cochran Q test (P > 0.050)". However, in the abstract one can find "The Barrett Universal II formula resulted ... in a higher percentage of eyes with prediction errors within ±0.50 D, ±0.75 D, and ±1.00 D." The same in P8 L163

- P8 L164 - present what test was used and the p values.

- P12 L238 as 20 eyes had a shallow ACD, how is it possible that 36.8% eyes (and not 35% or 40%) in this group had a prediction error of Â±0.50 D with the Hoffer Q formula

- Figure 3 is difficult analyze. Please consider a more various grayshade scheme or preferably adding colours...

The main limitation of the study is the group size, which is 219 eyes, compared to e.g., 18 501 in the study by Melles et al. Moreover, the novelty of this study is limited. Finally, the conclusions do not reflect the actual findings of the study (low difference between formulas, limited statistical significance)

I would consider publication after the revision.

Reviewer #4: This study evaluated the accuracy of the five intraocular lens (IOL) formulas integrated to a swept-source optical biometer. The manuscript was well written. I would like to add some comments.

1. Page 5, Line 108-111

Provide a reference for the criteria of subgroup classification as follows:

For the subgroup analysis, the AL, preoperative mean keratometry, and ACD (measured

from epithelium to lens) were divided into three subgroups: short (<22.5 mm), medium

(22.5–25.0 mm), and long (>25.0 mm); flat (<42.0 D), medium (42.0–45.0 D), and steep

(>45.0D); shallow (<2.50 mm), medium (2.5–3.5 mm), and deep (>3.5 mm).

2. Page 7, Line 150, Table 1

Please present the mean±SD of each subgroup in AL, K, and ACD.

3. Page 21, Fig.2.

For the MAE, we recommend the box whiskers plot instead of the line. The box whiskers plot can provide more information to the reader.

6. PLOS authors have the option to publish the peer review history of their article (what does this mean?). If published, this will include your full peer review and any attached files.

Reviewer #1: Yes: Giacomo Savini, MD

Reviewer #2: No

Reviewer #3: Yes: Piotr Kanclerz, MD, PhD

Reviewer #4: No

---

## [Author Response · Author response to Decision Letter 0]

29 Oct 2019

September 30, 2019

Editor in Chief 

PLOS ONE

Thank you very much for your letter regarding our revised manuscript. We revised the manuscript as recommended by the editor and reviewer. Each of the coauthors has seen and agrees with each of the changes made to the manuscript. Our responses to the comments of the editor and reviewer, including the changes made in the manuscript, are provided in a point-by-point manner in the following pages.

We hope that you will find the manuscript suitable for publication in your esteemed journal.

Yours sincerely,

Ji Won Jung, MD, PhD

Department of Ophthalmology, Inha University Hospital, 7-206, 3-ga,

Shinheung-dong, Jung-gu, Incheon 400-711, South Korea 

Tel.: 82-32-890-2400, Fax: 82-32-890-2403, E-mail: panch325@gmail.com

Reviewer Comments:

Reviewer #1: This is an interesting study. Although it does not add significant information to the recently published studies on much larger samples, it is the one of the first multiformula comparisons carried out with measurements by the IOLMaster 700.

Please find below some suggestions to improve the paper.

Thank you for your comments. We have revised this manuscript according to your suggestions.

METHODS

• Subgroup analysis: short eyes are usually defined as those with AL <22.0 mm (not 22.5 mm). Please correct and repeat analysis.

We agree with the reviewer’s opinion and we added the samples (from 219 eyes to 324 eyes) and repeated the analysis. The results revealed statistically significant differences, unlike previous version with borderline statistical significance. 

We also re-defined the subgroup classification. However, some of the subgroups contained only few subjects; therefore, we removed the subgroup analysis.

• Statistical analysis: in addition to Friedman’s test (to analyze the absolute prediction errors), I would include Repeated Measures ANOVA to compare the arithmetical prediction errors and their variance. This can be useful for both the whole sample and the different subgroups (shallow, medium and deep ACD, etc.), where it may reveal statistically significant differences among the 5 formulas.

We agree with the reviewer’s opinion. We added the samples (from 219 eyes to 324 eyes) and repeated the analysis. Friedman’s and Cochran’s Q test results revealed statistically significant differences among the five formulae, unlike previous version with borderline statistical significance.

• Statistical analysis: since you correctly used it, please cite Cochran’s Q test in the Methods section too.

We added Cochran’s Q test to the Methods section. 

• Please include errata for the Hoffer Q formula in the references (J Cataract Refract Surg 1993;19:700-712; errata, 1994;20:677; 2007;33:2-3.)

We added the errata in the reference 5. 

RESULTS

• It is quite hard to believe the Cochran’s Q test was not able to detect any statistically significant difference among the 5 formulas, as – for example – the Hoffer Q achieved only 69.6% of eyes within 0.5D, whereas the Barrett achieved 78.1% with the same prediction error. Please check carefully the statistical analysis once more.

We agree with the reviewer’s opinion. We added the samples and repeated the analysis. Cochran’s Q test results revealed statistically significant differences in the percentages of eyes within ±0.50, ±0.75 D, and ±1.00 D of the prediction errors among the five formulae. 

DISCUSSION

• Barrett formula does not belong to 5th generation formulas. Although unpublished and thus largely unknown, it should be classified among 4th generation ones.

We agree with the reviewer’s opinion and have made the recommended correction.

• Please compare your results with those reported by Hoffer Q when comparing the IOLMaster 700 to the Lenstar (JCRS 2016;42:1165-1172). They included the Hoffer Q, Holladay 1 and SRK/T formulas and achieved better results. It would be opportune to elaborate on the possible causes of this discrepancy.

As you suggested, we added this contents in Discussion section. 

• The whole discussion should be reviewed after ANOVA has been applied to the arithmetic prediction errors, as statistically significant differences may arise.

We agree with the reviewer’s opinion. We added the samples and repeated the analysis.

Friedman’s and Cochran’s Q test results revealed statistically significant differences among the five formulae. We have reviewed these results in the Discussion section.

Reviewer #2: Good results that support the conclusion. Reasonable statistical analysis. Correct terminology used. The ideas are clearly expressed in English. A bigger sample size is suggested in future research.

We agree with the reviewer’s opinion. We added the samples and repeated the analysis.

Reviewer #3: I have read the article by Kim et al. The authors have analyzed the accuracy of IOL power calculation formulas using the IOL Master 700.

Comments:

- P3 L53 - what about LCOR?

We agree with the reviewer’s opinion and we added the optical low-coherence reflectometry in P3 L53. 

- P3 L56 - some formulas also use preoperative refraction and age. Moreover, you should state that increasing the number, from the basic which is AL and K

We agree with the reviewer’s opinion and have made the suggested correction in P3 L56. 

- P3 L63 - be more precise, what formulas in what axial lengths? Is it concordant with the results of your study?

We agree with the reviewer’s opinion and we added these contents in P3. However, our data could not conclude because of the limited number of subjects in each subgroup. We plan to further evaluate in short or long AL subgroups. 

- P5 L108 - short eyes were determined as those having AL<22.5 mm, while long as those having 25.0 mm. As several studies used different cut-off values i.e. 22.0 mm or 26.0 mm choosing these values should be thoroughly explained.

- the group size for some subgroups (e.g. long eyes=16, flat keratometry=16, shallow ACD=20) very small, making an appropriate analysis difficult.

We agree with the reviewer’s opinion and we re-defined subgroup criteria. We reported the proportion of subgroup in order to understand our subjects in Table 1. However, because the sample sizes of new subgroups were small, therefore we have removed the subgroup analysis. The results of the entire group were similar to those of normal range subgroup, and the results of some subgroups with unusual range of ocular dimensions were difficult to draw conclusion because of small sample size. 

- in the results section you state: "The percentages of eyes within the prediction errors of ±0.25, ±0.50, ±0.75 D, and Â±1.00 D were not significantly different among the five formulas using the Cochran Q test (P > 0.050)". However, in the abstract one can find "The Barrett Universal II formula resulted ... in a higher percentage of eyes with prediction errors within ±0.50 D, ±0.75 D, and ±1.00 D." The same in P8 L163

- P8 L164 - present what test was used and the p values.

We agree with the reviewer’s opinion. We added the samples and repeated the analysis.

Cochran’s Q test revealed statistically significant differences among the five formulae. We revised these results in the Results and Discussion sections. 

- P12 L238 as 20 eyes had a shallow ACD, how is it possible that 36.8% eyes (and not 35% or 40%) in this group had a prediction error of Â±0.50 D with the Hoffer Q formula

This calculation was in error and has been corrected in the revised manuscript. 

- Figure 3 is difficult analyze. Please consider a more various grayshade scheme or preferably adding colours...

We corrected the figure by adding the colors. 

The main limitation of the study is the group size, which is 219 eyes, compared to e.g., 18 501 in the study by Melles et al. Moreover, the novelty of this study is limited. Finally, the conclusions do not reflect the actual findings of the study (low difference between formulas, limited statistical significance)

I would consider publication after the revision.

We agree with the reviewer’s opinion. We added the samples and repeated the analysis. Friedman’s and Cochran’s Q test results revealed statistically significant differences among the five formulae, unlike previous version with borderline statistical significance. We reviewed these results in Results and Discussion section. Our study had much smaller sample size than that of study by Melles et al. However, our study may be meaningful to compare the accuracy of various IOL formulae integrated to a swept-source optical biometer, the IOLMaster 700 for one popular IOL type (ZCB00). Please consider the revision of our manuscript. 

Reviewer #4: This study evaluated the accuracy of the five intraocular lens (IOL) formulas integrated to a swept-source optical biometer. The manuscript was well written. I would like to add some comments.

1. Page 5, Line 108-111

Provide a reference for the criteria of subgroup classification as follows:

For the subgroup analysis, the AL, preoperative mean keratometry, and ACD (measured

from epithelium to lens) were divided into three subgroups: short (<22.5 mm), medium

(22.5–25.0 mm), and long (>25.0 mm); flat (<42.0 D), medium (42.0–45.0 D), and steep

(>45.0D); shallow (<2.50 mm), medium (2.5–3.5 mm), and deep (>3.5 mm).

2. Page 7, Line 150, Table 1

Please present the mean±SD of each subgroup in AL, K, and ACD.

We agree with the reviewer’s opinion and have re-defined the subgroup criteria according to other references. However, because some subgroups were so small in size, we removed the subgroup analysis. 

3. Page 21, Fig.2.

For the MAE, we recommend the box whiskers plot instead of the line. The box whiskers plot can provide more information to the reader.

We agree with the reviewer’s opinion and we revised the figure as a box plot.

---

## [Decision Letter · Decision Letter 1]

26 Nov 2019

PONE-D-19-21173R1

Accuracy of intraocular lens power calculation formulas using a swept-source optical biometer

PLOS ONE

Dear Dr Jung,

Thank you for submitting your manuscript to PLOS ONE. After careful consideration, we feel that it has merit but does not fully meet PLOS ONE’s publication criteria as it currently stands. Therefore, we invite you to submit a revised version of the manuscript that addresses the points raised during the review process.

We would appreciate receiving your revised manuscript by Jan 10 2020 11:59PM. To enhance the reproducibility of your results, we recommend that if applicable you deposit your laboratory protocols in protocols.io, where a protocol can be assigned its own identifier (DOI) such that it can be cited independently in the future. For instructions see: http://journals.plos.org/plosone/s/submission-guidelines#loc-laboratory-protocols

We look forward to receiving your revised manuscript.

Kind regards,

Ireneusz Grulkowski, PhD

Academic Editor

PLOS ONE

Additional Editor Comments (if provided):

Some minor editing of the plot needed as given by the Reviewer 4

Reviewers' comments:

Reviewer's Responses to Questions

**Comments to the Author**

1. If the authors have adequately addressed your comments raised in a previous round of review and you feel that this manuscript is now acceptable for publication, you may indicate that here to bypass the “Comments to the Author” section, enter your conflict of interest statement in the “Confidential to Editor” section, and submit your "Accept" recommendation.

Reviewer #1: All comments have been addressed

Reviewer #2: All comments have been addressed

Reviewer #4: All comments have been addressed

2. Is the manuscript technically sound, and do the data support the conclusions?

Reviewer #1: Yes

Reviewer #2: Yes

Reviewer #4: Yes

3. Has the statistical analysis been performed appropriately and rigorously? 

Reviewer #1: Yes

Reviewer #2: Yes

Reviewer #4: Yes

4. Have the authors made all data underlying the findings in their manuscript fully available?

Reviewer #1: Yes

Reviewer #2: Yes

Reviewer #4: Yes

5. Is the manuscript presented in an intelligible fashion and written in standard English?

Reviewer #1: Yes

Reviewer #2: (No Response)

Reviewer #4: Yes

6. Review Comments to the Author

Reviewer #1: Just a minor comment: in table 1, medium axial length is still defined as between 22.5 and 26 mm. Please correct 22.5 and change it into 22

Reviewer #2: Good results that support the conclusion. Reasonable statistical analysis. Correct terminology used. The ideas are clearly expressed in English. A bigger sample size is suggested in future research.

Reviewer #4: We have confirmed the corrections in the manuscript.

Please indicate the minimum and maximum values for the box & whisker plots.

7. PLOS authors have the option to publish the peer review history of their article (what does this mean?). If published, this will include your full peer review and any attached files.

Reviewer #1: No

Reviewer #2: No

Reviewer #4: No

---

## [Author Response · Author response to Decision Letter 1]

29 Nov 2019

Reviewer Comments:

Reviewer #1: Just a minor comment: in table 1, medium axial length is still defined as between 22.5 and 26 mm. Please correct 22.5 and change it into 22

Thank you for your comment. We corrected it. 

Reviewer #2: Good results that support the conclusion. Reasonable statistical analysis. Correct terminology used. The ideas are clearly expressed in English. A bigger sample size is suggested in future research.

Thank you for your comment. We added the need of bigger sample size in future research in Discussion section. 

Reviewer #4: We have confirmed the corrections in the manuscript.

Please indicate the minimum and maximum values for the box & whisker plots.

Thank you for your comment. We added the minimum and maximum values in Fig2.

---

## [Decision Letter · Decision Letter 2]

26 Dec 2019

Accuracy of intraocular lens power calculation formulas using a swept-source optical biometer

PONE-D-19-21173R2

Dear Dr. Jung,

We are pleased to inform you that your manuscript has been judged scientifically suitable for publication and will be formally accepted for publication once it complies with all outstanding technical requirements.

With kind regards,

Ireneusz Grulkowski, PhD

Academic Editor

PLOS ONE

Additional Editor Comments (optional):

Reviewers' comments:

Reviewer's Responses to Questions

**Comments to the Author**

1. If the authors have adequately addressed your comments raised in a previous round of review and you feel that this manuscript is now acceptable for publication, you may indicate that here to bypass the “Comments to the Author” section, enter your conflict of interest statement in the “Confidential to Editor” section, and submit your "Accept" recommendation.

Reviewer #4: All comments have been addressed

2. Is the manuscript technically sound, and do the data support the conclusions?

Reviewer #4: Yes

3. Has the statistical analysis been performed appropriately and rigorously? 

Reviewer #4: Yes

4. Have the authors made all data underlying the findings in their manuscript fully available?

Reviewer #4: Yes

5. Is the manuscript presented in an intelligible fashion and written in standard English?

Reviewer #4: Yes

6. Review Comments to the Author

Reviewer #4: The manuscript has been modified according to the requirements. This manuscript is expected to give readers useful information.

7. PLOS authors have the option to publish the peer review history of their article (what does this mean?). If published, this will include your full peer review and any attached files.

Reviewer #4: No

---

## [Editor Report · Acceptance letter]

30 Dec 2019

PONE-D-19-21173R2 

Accuracy of intraocular lens power calculation formulas using a swept-source optical biometer 

Dear Dr. Jung:

I am pleased to inform you that your manuscript has been deemed suitable for publication in PLOS ONE. Congratulations! Your manuscript is now with our production department. 

With kind regards,

on behalf of

Dr. Ireneusz Grulkowski 

Academic Editor

PLOS ONE